# CoBA: Causal Contextual Bandits with Active Data Integration

## Abstract

We study a contextual bandit setting where the agent has the ability to request multiple data samples – corresponding to potentially different context-action pairs – simultaneously in one-shot within a budget, along with access to causal side information. This new formalism provides a natural model for several real-world scenarios where parallel targeted experiments can be conducted. We propose a new algorithm that utilizes a novel entropy-like measure that we introduce. We perform multiple experiments, both using purely synthetic data and using a real-world dataset, and show that our algorithm performs better than baselines in all of them. In addition, we also study sensitivity of our algorithm's performance to various aspects of the problem setting. We also show that the algorithm is sound; that is, as budget increases, the learned policy eventually converges to an optimal policy. Further, we show a bound on its regret under additional assumptions. Finally, we study fairness implications of our methodology.

## 1 Introduction

Learning to make decisions that depend on context has a wide range of applications – software product experimentation, personalized medical treatments, recommendation systems, marketing campaign design, etc. Contextual bandits (Lattimore & Szepesvári, 2020) have been used to model such problems with good success (Liu et al., 2018; Sawant et al., 2018; Bouneffouf et al., 2020; Ameko et al., 2020).

Contextual bandits have been studied in two primary variants – interactive (e.g., Agarwal et al. (2014); Dimakopoulou et al. (2019)) and offline (e.g., Swaminathan & Joachims (2015a); Li et al. (2015)). In the former, the agent repeatedly interacts with the environment (observe context, choose action, receive reward) and updates its internal state after every interaction, whereas in the latter the agent is provided an offline log of data to learn from. The objective in both cases is to learn a near-optimal policy that maps contexts to actions. Interactive bandits are favored in applications where interventions are cheap, such as in software product experimentation (e.g., Optimizely (2023)). On the other hand, offline contextual bandits have become increasingly popular in scenarios where interactions are costly to actualize in the real world (e.g., conducting physical experiments) or prohibited (e.g., in the healthcare domain with human subjects).

While offline contextual bandit algorithms do provide methods to utilize existing data to learn policies, it is an unreasonably constrained model for many problems; often in the real world it is possible to *acquire additional data in one-shot* – but at a cost and within a budget. To the best of our knowledge, there has not been any investigation on what the best way is to actively obtain additional experimental data in one shot and incorporate it with aim of learning a good policy.[1]

### 1.1 Motivating examples

In software product development, product teams are frequently interested in learning the best software variant to show or roll out to each subgroup of users. To achieve this, it is often possible to conduct

---

[1]Note that this is fundamentally different from interactive bandit settings because the agent cannot iteratively acquire samples as it updates its knowledge, but rather raise a one-shot data request and learn from it. This is more reflective of many real world scenarios.

targeting at scale simultaneously for various combinations of contexts (representing user groups) and actions (representing software variants) – instead of one experiment at a time – by routing traffic appropriately (see Google (2021) for an example). These targeted experiments[2] can be used to compute relevant metrics (e.g., click-through rate) for each context-action pair. Further, we might have some qualitative domain knowledge of how some context variables are causally related to others; for example, we might know that `os` has a causal relationship to `browser`; we would like to exploit this knowledge to learn better policies. This can be naturally modeled as a question of "what *table* of targeted experimental data needs to be acquired and how to integrate that data, so as to learn a good policy?"; here the table's rows and columns are contexts and actions, and each cell specifies the number of data samples (zero or more) required corresponding to the context-action pair.

As another example, in ads targeting (e.g., Google (2023)) the objective is to learn the best ads to show each group of people. This is done by conducting experiments where different groups of people are shown different ads simultaneously. Further, letting context variables model the features of groups, we might have some knowledge about how some of the features are causally related to others; for example, we might know that `country` causally affects `income`. Our framework provides a natural model for this setting. Further, we might also be interested in ensuring that the ads meet certain criteria for fairness. For example, there might be some variables such as `race` that could be sensitive from a fairness perspective, and we would not want the agent to learn policies that depends on these variables. We discuss fairness implications of our algorithm in Section 6. These are just two of many scenarios where this framework provides a natural model. Two additional examples include experimental design for marketing insights (e.g., Persado (2023)) and recommendation systems (e.g., ScaleAI (2023)).

## 1.2   Our framework

Our framework captures the various complexities and nuances described in the examples in Section 1.1. We present an overview here; please see Section 3 for the mathematical formalism. At the start, the agent is given a (possibly empty) log of offline data – consisting of context-action-reward tuples — generated from some unknown policy. The context variables are partitioned into two sets – the main set and the auxiliary set (possibly empty). The agent observes all context variables during training, but learns a policy that only depends on the main set of context variables; this also provides a way to ensure that the learned policy meets certain definitions of fairness (see Section 6 for a more detailed discussion). Further, the agent also has some qualitative[3] causal side-information available, likely from domain knowledge. This causal side-information is encoded as a causal graph between contextual variables. A key implication of the causal graph is information leakage (Lattimore et al., 2016; Subramanian & Ravindran, 2022) – getting samples for one context-action pair provides information about other context-action pairs because of shared pathways in the causal graph.

Given the logged data and the causal graph, the agent's problem is to decide the set of targeted experimental samples to acquire within a budget, and then integrate the returned samples. More specifically, the agent is allowed to make a *one-shot* request for data in the form of a table specifying the number of samples it requires for each context-action pair, subject to the total cost being within the budget. The environment then returns the requested samples after conducting the targeted interventions, and the agent integrates those samples to update its internal beliefs and learned policy; this constitutes the training phase of the agent. The core problem of the agent is to choose these samples in a way that straddles the trade-off between choosing more samples for context-action pairs it knows is likely more valuable (given its beliefs) and choosing more samples to explore less-seen context-action pairs – while taking into account the budget and the information leakage from the causal graph. After training, the agent moves to an inference phase, where it returns an action (according to the learned policy) for every context it encounters.

---

[2]Each of these experiments is a "targeted intervention", formalized in Subramanian & Ravindran (2022) as an intervention targeted on a subgroup specified by a particular assignment of values to the context variables.

[3]By qualitative, we mean that the agent can know the causal *graph*, but not the conditional probability distributions of the variables. See Section 3 for a more detailed discussion.

### 1.3 Contributions

1. This is the first work to study how to *actively obtain and integrate* a table of *multiple samples in one-shot* in a contextual bandit setting. Further, we study this in the presence of a causal graph, making it one of the very few works to study *integration of causal side-information* by contextual bandit agents. See Section 2 for a more detailed discussion on related work, and Section 3 for the mathematical formalism of the problem.

2. We propose a novel algorithm (Section 4.3) that works by minimizing a new entropy-like measure called $\Upsilon(.)$ that we introduce. See Section 4 for a full discussion on the approach. We also show that the method is sound – that is, as the budget tends to infinity, the algorithm's regret converges to 0 (Section 4.4). We also provide a regret bound, under some additional assumptions (Section 4.5).

3. We show results of experiments, using purely synthetically generated data and an experiment inspired by real-world data, that demonstrate that our algorithm performs better than baselines. We also study sensitivity of the results to key aspects of the problem setting. Refer Section 5.

4. We discuss fairness implications of our method, and show that it achieves counterfactual fairness (Section 6).

## 2 Related work

Causal bandits have been studied in the last few years (e.g., Lattimore et al. (2016); Yabe et al. (2018); Lu et al. (2020)), but they study this in a multi-armed bandit setting where the problem is identification of one best action. There is only one work (Subramanian & Ravindran, 2022) studying causal *contextual* bandits – where the objective is to learn a *policy* mapping contexts to actions – and this is the closest related work. While we do leverage some ideas introduced in that work in our methodology and in the design of experiments, our work differs fundamentally from this work in important ways. Subramanian & Ravindran (2022) consider a standard interactive setting where the agent can repeatedly act, observe outcomes and update its beliefs, whereas in our work the agent has a one-shot data collection option for samples from *multiple* context-action pairs. This fundamentally changes the nature of the optimization problem as we will see in Section 4; it also makes it a more natural model in a different set of applications, some of which were discussed in Section 1.1. Further, our work allows for arbitrary costs for collecting those samples, whereas they assume every intervention is of equal cost.

Contextual bandits in purely offline settings, where decision policies are learned from logged data, is a well-studied problem. Most of the work involves inverse propensity weighting based methods (such as Swaminathan & Joachims (2015a;b); Joachims et al. (2018)). Contextual bandits are also well-studied in purely interactive settings (see Lattimore & Szepesvári (2020) for a discussion on various algorithms). However, in contrast to our work, none of these methods can integrate causal side information or provide a way to optimally acquire and integrate new data. Further, none of these methods study actively obtaining and integrating a table of data containing samples corresponding to multiple context-action pairs.

Active learning (Settles, 2009) studies settings where an agent is allowed to query an oracle for ground truth labels for certain data points. This has been studied in supervised learning settings where the agent receives ground truth feedback; in contrast, in our case, the agent receives outcomes only for actions that were taken ("bandit feedback"). However, despite this difference, our approach can be viewed as incorporating some elements of active learning into contextual bandits by enabling the agent to acquire additional samples at a cost. There has been some work that has studied contextual bandits with costs and budget constraints (e.g., Agrawal & Goyal (2012); Wu et al. (2015)). There has also been work that has explored contextual bandit settings where the agent can not immediately integrate feedback from the environment, but can do so only in batches (Zhang et al., 2022; Ren et al., 2022; Han et al., 2020). However, all these works consider settings with repeated interactions, whereas our work considers a one-shot setting where the agent chooses multiple context-action pairs simultaneously. Further, none of these works provide a way to integrate causal side information.

Table 1: Summary of key notation

| Notation | Meaning |
|---|---|
| $X$ | action variable |
| $Y$ | reward variable |
| $\mathcal{C}^A, \mathcal{C}^B$ | set of main context variables and set of auxiliary context variables, respectively; so the set of all context variables is $\mathcal{C} = \mathcal{C}^A \cup \mathcal{C}^B = \{..., C_i, ...\}$. |
| Capital letters | a random variable; e.g., $C_1$ or $X$ |
| Small letters | a random variable's value; e.g., $c_1$ or $x$ |
| Small bold font | an assignment of values to a set of random variables; for example, $\mathbf{c}$ denotes a specific choice of values taken by variables in $\mathcal{C}$ |
| $\hat{\mathbb{P}}, \hat{\mathbb{E}}$ | estimate of distribution $\mathbb{P}$ and expectation $\mathbb{E}$ based on current beliefs |
| $\mathsf{val}(V), \mathsf{val}(\mathcal{V})$ | set of values taken by the variable $V$, and set of variables $\mathcal{V}$, respectively. |
| $\hat{\phi}, \phi^*$ | the learned policy and an optimal policy, respectively |
| $\Upsilon$ | entropy-like measure used in our algorithm; defined in Equation 3 |
| $\mathbf{pa}_V$ | value of variables in $PA_V$, the parents of $V$ |
| $N_{x,\mathbf{c}^A}$ | number of samples requested corresponding to $X = x$ and $\mathcal{C}^A = \mathbf{c}^A$ |
| $\beta(x, \mathbf{c}^A, N_{x,\mathbf{c}^A})$ | cost of acquiring $N_{x,\mathbf{c}^A}$ samples corresponding to $X = x$ and $\mathcal{C}^A = \mathbf{c}^A$ |
| $B$ | budget |
| $\mathbf{a}\langle \mathcal{B} \rangle$ | if $\mathbf{a}$ is an assignment of values to $\mathcal{A}$, then $\mathbf{a}\langle \mathcal{B} \rangle$ is assignment of those values to respective variables $\mathcal{B}$; $\mathbf{a}\langle \mathcal{B} \rangle = \emptyset$ if $\mathcal{A} \cap \mathcal{B} = \emptyset$. |

## 3 Problem formalism

**Underlying model**  We model the underlying environment as a causal model $\mathcal{M}$, which is defined by a directed acyclic graph $\mathcal{G}$ over all variables (the "causal graph") and a joint probability distribution $\mathbb{P}$ that factorizes over $\mathcal{G}$ (Pearl, 2009b; Koller & Friedman, 2009). The set of variables in $\mathcal{G}$ consists of the action variable ($X$), the reward variable ($Y$), and the set of context variables ($\mathcal{C}$). Each variable takes on a finite, known set of values; note that this is quite general, and accommodates categorical variables. $\mathcal{C}$ is partitioned into the set of main context variables ($\mathcal{C}^A$) and the set of (possibly empty) auxiliary context variables ($\mathcal{C}^B$). That is, $\mathcal{C} = \mathcal{C}^A \cup \mathcal{C}^B$.

The agent knows only $\mathcal{G}$ but not $\mathcal{M}$; therefore, the agent has no *a priori* knowledge of the conditional probability distributions (CPDs) of the variables.

**Protocol**  In addition to knowing $\mathcal{G}$, the agent also has access to logged offline data, $\mathcal{D}_L = \{(\mathbf{c}_i, x_i, y_i)\}$, where each $(\mathbf{c}_i, x_i, y_i)$ is sampled from $\mathcal{M}$ and $x_i$ is chosen following some unknown policy. Unlike many prior works, such as Swaminathan & Joachims (2015a), the agent here does *not* have access to the logging propensities.

The agent then specifies in one shot the number of samples $N_{x,\mathbf{c}^A}$ it requires for each pair $(x, \mathbf{c}^A)$. We denote the full table of these values by $\mathbf{N} \triangleq \bigcup_{x,\mathbf{c}^A}\{N_{x,\mathbf{c}^A}\}$. Given a $(x, \mathbf{c}^A)$, there is an arbitrary cost $\beta(x, \mathbf{c}^A, N_{x,\mathbf{c}^A})$ associated with obtaining those samples. The total cost should be at most a budget $B$. For each $(x, \mathbf{c}^A)$, the environment returns $N_{x,\mathbf{c}^A}$ samples of the form $(\mathbf{c}^B, y) \sim \mathbb{P}(\mathcal{C}^B, Y \mid do(x), \mathbf{c}^A)$.[4] Let's call this acquired dataset $\mathcal{D}_A$. The agent utilizes $\mathcal{D}_A$ along with $\mathcal{D}_L$ to learn a good policy.

**Objective**  The agent's objective is to learn a policy $\hat{\phi} : \mathsf{val}(\mathcal{C}^A) \to \mathsf{val}(X)$ such that expected simple regret is minimized:

$$Regret \triangleq \sum_{\mathbf{c}^A} [\mu^*_{\mathbf{c}^A} - \hat{\mu}_{\mathbf{c}^A}] \cdot \mathbb{P}(\mathbf{c}^A)$$

---

[4]See Pearl (2009b; 2019) for more discussion on the *do*() operation.

where $\phi^*$ is an optimal policy, $\mu^*_{\mathbf{c}^A} \triangleq \mathbb{E}[Y|do(\phi^*(\mathbf{c}^A), \mathbf{c}^A)]$ and $\hat{\mu}_{\mathbf{c}^A} \triangleq \mathbb{E}[Y|do(\hat{\phi}(\mathbf{c}^A), \mathbf{c}^A)]$.

Table 1 provides a summary of the key notation used in this paper.

### 3.1 Assumptions

We assume that $X$ has exactly one outgoing edge, $X \to Y$, in $\mathcal{G}$. This is suitable to express a wide range of problems such as personalized treatments or software experimentation where the problem is to learn the best action under a context, but the action or treatment does not affect context variables. We also make a commonly-made assumption (see Guo et al. (2020)) that there are no unobserved confounders. Similar to Subramanian & Ravindran (2022), we make an additional assumption that simplifies the factorization in Section 4.2: $\{C \text{ confounds } C' \in \mathcal{C}^A \text{ and } Y\} \implies C \in \mathcal{C}^A$; a *sufficient* condition for this to be true is if $\mathcal{C}^A$ is ancestral.[5] This last assumption is a simplifying assumption and can be relaxed in the future.

## 4 Approach

### 4.1 Overall idea

In our approach, the agent works by maintaining beliefs regarding every conditional probability distribution (CPD). It first uses $\mathcal{D}_L$ to update its initial CPD beliefs; this, in itself, makes use of information leakage provided the causal graph. It next needs to choose $\mathcal{D}_A$, which is the core problem. The key tradeoff facing the agent is the following: it needs to choose between allocating more samples to context-action pairs that it believes are more valuable and to context-action pairs that it knows less about. Unlike Subramanian & Ravindran (2022), it cannot interactively choose and learn, but instead has to choose the whole $\mathcal{D}_A$ in one shot – necessitating the need to account for multiple overlapping information leakage pathways resulting from the multitude of samples. In addition, these samples have a cost to acquire, given by an arbitrary cost function, along with a total budget.

**Towards solving this**  To achieve this, we define a novel function $\Upsilon(\mathbf{N})$ that captures a measure of overall entropy weighted by value. The idea is that minimizing $\Upsilon$ results in a good policy; that is, the agent's problem now becomes that of minimizing $\Upsilon$ subject to budget constraints. In Section 4.2, we formally define $\Upsilon(\mathbf{N})$ and provide some intuition. Later, we provide experimental support (see Section 5), along with some theoretical grounding to this intuition (see Theorem 4.1).

### 4.2 The optimization problem

Determine $N_{x,\mathbf{c}^A}$ for each $(x, \mathbf{c}^A)$ such that

$$\Upsilon(\mathbf{N})$$

is minimized, subject to

$$\sum_{x,\mathbf{c}^A} \beta(x, \mathbf{c}^A, N_{x,\mathbf{c}^A}) \leq B$$

where $\mathbf{N} \triangleq \bigcup_{x,\mathbf{c}^A} \{N_{x,\mathbf{c}^A}\}$. We will next define $\Upsilon(\mathbf{N})$.

**Defining the objective function $\Upsilon(\mathbf{N})$**  The conditional distribution $\mathbb{P}(V|\mathbf{pa}_V)$ for any variable $V$ is modeled as a categorical distribution whose parameters are sampled from a Dirichlet distribution (the belief distribution). That is, $\mathbb{P}(V|\mathbf{pa}_V) = \mathsf{Cat}(V; b_1, ..., b_r)$, where $(b_1, ..., b_r) \sim \mathsf{Dir}(\theta_{V|\mathbf{pa}_V})$, and $\theta_{V|\mathbf{pa}_V}$ is a vector denoting the parameters of the Dirichlet distribution.

Actions in a contextual bandit setting can be interpreted as $do()$ interventions on a causal model (Zhang & Bareinboim, 2017; Lattimore et al., 2016). Therefore, the reward $Y$ when an agent chooses action $x$ against context $\mathbf{c}^A$ can be thought of as being sampled according to $\mathbb{P}[Y|do(x), \mathbf{c}^A]$. Under the assumptions

---

[5]That is, if $\mathcal{C}^A$ contains all its ancestors.

described in Section 3.1, we can factorize as follows:

$$\mathbb{E}[Y|do(x), \mathbf{c}^A] = \sum_{\mathbf{c}^B \in \mathsf{val}(\mathcal{C}^B)} \left[ \mathbb{P}(Y = 1|x, \mathbf{c}\langle PA_Y \rangle) \prod_{c \in \mathbf{c}^B} \mathbb{P}(C = c|\mathbf{c}\langle PA_C \rangle) \right] \tag{1}$$

Note that our beliefs about each CPD in Equation 1 are affected by samples corresponding to multiple $(x, \mathbf{c}^A)$. To capture this, we construct a CPD-level uncertainty measure which we call $Q(.)$:

$$Q(\mathbb{P}[V|\mathbf{pa}_V], \mathbf{N}) \triangleq \sum_{x, \mathbf{c}^A} \left( \frac{1}{1 + \ln(N_{x, \mathbf{c}^A} + 1)} \right) \mathsf{Ent}^{new}(\mathbb{P}[V|\mathbf{pa}_V]) \bigg|_{x\langle PA_V \rangle = \mathbf{pa}_V \langle X \rangle, \;\; \mathbf{c}^A \langle PA_V \rangle = \mathbf{pa}_V \langle \mathcal{C}^A \rangle} \tag{2}$$

Here $\mathsf{Ent}^{new}$ is defined in the same way as in Subramanian & Ravindran (2022); for reference, we reproduce the definition in Appendix A. Finally, we construct $\Upsilon(\mathbf{N})$ as:

$$\Upsilon(\mathbf{N}) \triangleq \sum_{x, \mathbf{c}} \left[ \left[ \sum_{V \in \mathcal{C}^B \cup \{Y\}} Q(\mathbb{P}[V|\mathbf{c}\langle PA_V \rangle], \mathbf{N}) \right] \cdot \hat{\mathbb{P}}(\mathbf{c}) \cdot \hat{\mathbb{E}}[Y|x, \mathbf{c}\langle PA_Y \rangle] \right] \tag{3}$$

**Intuition behind $Q(.)$ and $\Upsilon(.)$** Intuitively, $\mathsf{Ent}^{new}$ provides a measure of entropy if *one* additional sample corresponding to $(x, \mathbf{c}^A)$ is obtained and used to update beliefs. $Q(.)$ builds on it and captures the fact that the beliefs regarding any CPD $\mathbb{P}[V|\mathbf{pa}_V]$ can be updated using information leakage[6] from samples corresponding to *multiple* $(x, \mathbf{c}^A)$; it does this by selecting the relevant $(x, \mathbf{c}^A)$ pairs making use of the causal graph $\mathcal{G}$ and aggregating them. In addition, $Q(.)$ also captures the fact that entropy reduces non-linearly with the number of samples. Finally, $\Upsilon(\mathbf{N})$ provides an aggregate (weighted) resulting uncertainty from choosing $N_{x, \mathbf{c}^A}$ samples of each $(x, \mathbf{c}^A)$. The weighting in $\Upsilon(.)$ provides a way for the agent to relatively prioritize context-action pairs that are higher-value according to its beliefs.

### 4.3 Algorithm

The full learning algorithm, which we call CoBA, is given as Algorithm 1a. After learning, the algorithm for inferencing on any test context (i.e., returning the action for the given context) is given as Algorithm 1b. The core problem (Step 2 in Algorithm 1a) is a nonlinear optimization problem with nonlinear constraints and integer variables. It can be solved using any of the various existing solvers. In our experiments in Section 5, we solve it approximately using the `scipy` Python library.

### 4.4 Soundness

As $B \to \infty$, the agent's regret will tend towards 0 (Theorem 4.1). It demonstrates the soundness of our approach by showing that as the budget increases, the learned policy will eventually converge to an optimal policy.

**Theorem 4.1** (Soundness). *As $B \to \infty$, Regret $\to 0$.*

The proof of Theorem 4.1 is presented in Appendix B.

### 4.5 Regret bound

The limiting case where $B \to \infty$ was already analyzed and we showed that our algorithm converges to an optimal policy in that case (Theorem 4.1). In Theorem 4.2, in contrast, we are interested in the finite-$B$ case. This is of interest in practical settings where the budget is usually small. We prove a regret bound under additional additional assumptions (A2) which we describe in Appendix C. Here define $m \triangleq \min_{x, \mathbf{c}^A} \pi(x|\mathbf{c}^A)$, where $\pi$ is the (unknown) logging policy that generated $\mathcal{D}_L$; and $M_\mathcal{V} \triangleq |\mathsf{val}(\mathcal{V})|$.

---

[6]Information leakage arises from shared pathways in $\mathcal{M}$; or equivalently, due to shared CPDs in the factorization of $\mathbb{P}$.

---

**Algorithm 1a:** Learning phase of CoBA

    **Data:** Causal graph $\mathcal{G}$; initial dataset $\mathcal{D}_L$

    **Initialization:** For all $V \in \mathcal{C} \cup \{Y\}$ and for all $\mathbf{pa}_V$, set $\theta_{V|\mathbf{pa}_V} = (1, ..., 1)$.

**1** Update all beliefs using initial dataset $\mathcal{D}_L$ by calling **update_beliefs**$(\mathcal{D}_L)$

**2** Solve the following

$$\mathbf{N} = \arg\min_{\mathbf{N}'} \Upsilon(\mathbf{N}')$$

    subject to

$$\sum_{x,\mathbf{c}^A} \beta(x, \mathbf{c}^A, N'_{x,\mathbf{c}^A}) \leq B$$

**3** Place data request $\mathbf{N}$; environment returns dataset $\mathcal{D}_A$ as discussed in Section 3.

**4** Update all beliefs using $\mathcal{D}_A$ by calling **update_beliefs**$(\mathcal{D}_A)$

    **Result:** Final set of beliefs for all $V, \mathbf{pa}_V$: $\left\{..., \theta_{V|\mathbf{pa}_V}, ...\right\}$

**5** *Procedure* **update_beliefs**$(\mathcal{D})$

**6**     **for** *each sample* $(x, \boldsymbol{c}, y) \in \mathcal{D}$ **do**

**7**         let $\tilde{\mathbf{c}} \triangleq \mathbf{c} \cup \{x, y\}$

**8**         **for** *each* $V \in \mathcal{C} \cup \{Y\}$ **do**

**9**              $\theta_{V|\tilde{\mathbf{c}}\langle PA_V\rangle}\left[\tilde{\mathbf{c}}\langle V\rangle\right] \leftarrow \theta_{V|\tilde{\mathbf{c}}\langle PA_V\rangle}\left[\tilde{\mathbf{c}}\langle V\rangle\right] + 1$

**10**         **end**

**11**     **end**

---

**Algorithm 1b:** Inference phase

    **Data:** Causal graph $\mathcal{G}$, learned beliefs $\left\{..., \theta_{V|\mathbf{pa}_V}, ...\right\}$, test context $\mathbf{c}^A$

**1** **for** *every* $V$, $\mathbf{pa}_V$ **do**

**2**     **for** $v \in \mathsf{val}(V)$ **do**

**3**         Set $\hat{\mathbb{P}}(V = v | \mathbf{pa}_V) = \dfrac{\theta_{V|\mathbf{pa}_V}^{(v)}}{\sum_{v'} \theta_{V|\mathbf{pa}_V}^{(v')}}$

**4**     **end**

**5** **end**

**6** **for** $x \in \mathsf{val}(X)$ **do**

**7**     Compute $\hat{\psi}(x, \mathbf{c}^A) \triangleq \hat{\mathbb{E}}[Y | do(x), \mathbf{c}^A]$ using $\hat{\mathbb{P}}$ in Equation (1)

**8** **end**

    **Result:** Return $\hat{\phi}(\mathbf{c}^A) \triangleq \arg\max_x \hat{\psi}(x, \mathbf{c}^A)$

---

**Theorem 4.2** (Regret bound). *Under the additional assumptions (A2) mentioned in Appendix C, for any $0 < \delta < 1$, with probability $\geq 1 - \delta$,*

$$Regret \in O\left(|\mathcal{C}|\sqrt{\left(\frac{1}{mB - \epsilon}\right)\ln\frac{M_X M_{\mathcal{C}}}{\delta}}\right)$$

*where $\epsilon \in O\left(\sqrt{B \ln\left(M_X M_{PA_Y} M_{\mathcal{C}}/\delta\right)}\right)$, ignoring terms that are constant in $B$, $m$, $\delta$, $|\mathcal{C}|$ and the number of possible context-action pairs.*

The proof of Theorem 4.2 is presented in Appendix C. It closely follows the regret bound proof in Subramanian & Ravindran (2022) and adapts it to our setting. The purpose of the proof is to establish an upper bound on performance, and not to provide a tight bound. Bounding regret without these additional assumptions (A2) is left for future work.

# 5 Experimental results

## 5.1 Baselines and experimental setup

**Baselines**   There are no existing algorithms that directly map to our setting. Therefore, we construct a set of natural baselines and study the performance of our algorithm CoBA against them. EqualAlloc allocates an equal number of samples to all $(x, \mathbf{c}^A)$; this provides a good distribution of samples to all context-action pairs. MaxSum maximizes the *total* number of samples summed over all $(x, \mathbf{c}^A)$. PropToValue allocates a number of samples to $(x, \mathbf{c}^A)$ that is proportional to $\hat{\mathbb{P}}(\mathbf{c}^A) \cdot \hat{\mathbb{E}}[Y|do(x), \mathbf{c}^A]$; this allocates relatively more samples to context-action pairs that are more "valuable" based on the agent's current beliefs. All baselines first involve updating the agent's starting beliefs regarding the CPDs of $\mathcal{M}$ using $\mathcal{D}_L$ (same as Step 1 of Algorithm 1a) before allocating samples for active obtainment as detailed above. After $\mathcal{D}_A$ is returned by the environment, all baselines use it update their beliefs (same as Step 4 of Algorithm 1a).

**Experiments**   Similar to Subramanian & Ravindran (2022), we consider a causal model $\mathcal{M}$ whose causal graph $\mathcal{G}$ consists of the following edges: $C_1 \to C_0$, $C_0 \to X$, $C_0 \to Y$, $X \to Y$. We let $\mathcal{C}^A = \{C_1\}$ and $\mathcal{C}^B = \{C_0\}$. We use this causal graph for all experiments except Experiment 3, for which we use the graph shown in Figure 3a.

Experiments 1 and 2 analyze the performance of our algorithm in a variety of settings, similar to those used in Subramanian & Ravindran (2022). Experiment 3 analyzes the performance of the algorithm on a setting calibrated using real-world CRM sales-data provided in Subramanian & Ravindran (2022). The details of all the parameterizations are provided as part of the supplementary material (see Appendix G). Experiments 4 through 7 analyze sensitivity of our algorithm's performance to various aspects of the problem setting. In all experiments, except Experiment 6, we set the cost function $\beta(.)$ to be proportional to the number of samples – a natural definition of cost; in Experiment 6, we analyze sensitivity to cost function choice. Additional experiments providing more insights into *why* our algorithm performs better than baselines are discussed in Appendix E. Appendix D reports results of Experiment 1 and 2 for larger values of $B$ (until all algorithms converge), providing empirical evidence of our algorithm's improved asymptotic behavior.

**Remark**   If the specific parameterization of $\mathcal{M}$ were given *a priori*, it is possible to come up with an algorithm that performs optimally in that particular setting. However, the objective is to design a method that performs well *overall* without this *a priori* information. Consider the relative performance of the baselines in Experiments 2 and 3. We will see that while EqualAlloc performs better than MaxSum and PropToValue in Experiments 3 (Section 5.4), it performs worse than those two in Experiment 2 (Section 5.3). However, our algorithm performs better than all three baselines in all experiments, corroborating our algorithm's overall better performance.

## 5.2 Experiment 1 (representative settings)

Different parameterizations of $\mathcal{M}$ can produce a wide range of possible settings. Given this, the first experiment studies the performance of our algorithm over a set of "*representative settings*". Each of these settings has a natural interpretation; for example, $\mathcal{C}^A$ could represent the set of person-level features that we are learning a personalized treatment for, or it could represent the set of customer attributes over which we're learning a marketing policy. The settings capture the intuition that high-value contexts (contexts for which, if the optimal action is learned, high expected rewards accrue to the agent) occur relatively less frequently (say, 20% of the time), but that there can be variation in other aspects. Specifically, the variations come from the number of different values of $\mathbf{c}^A$ over which the 20% probability mass is spread, and in how "risky" a particular context is (e.g., difference in rewards between the best and worst actions). The full details of the parameterizations are provided as part of the supplementary material (refer Appendix G). The number of samples[7] in the initial dataset $\mathcal{D}_L$ is kept at $0.5 \cdot |\mathsf{val}(\mathcal{C}^A)| \cdot |\mathsf{val}(X)|$. In each run, the agent is presented with a randomly selected setting from the representative set. Results are averaged over 50 independent runs; error bars display $\pm 2$ standard errors.

---

[7]We consider a uniformly exploring logging policy for $\mathcal{D}_L$; that is, context variables for each sample are realized as per the natural distribution induced by $\mathcal{M}$, but $X$ is chosen randomly.

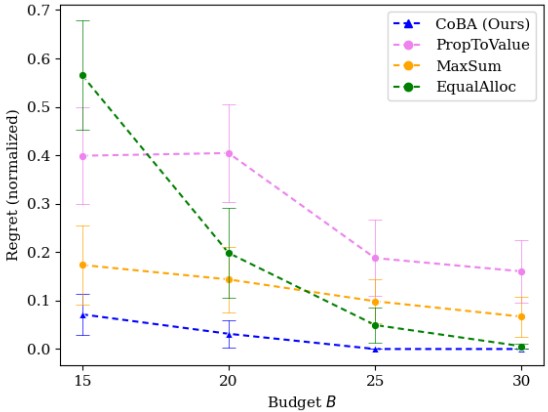

Figure 1: Experiment 1 results (Section 5.2).

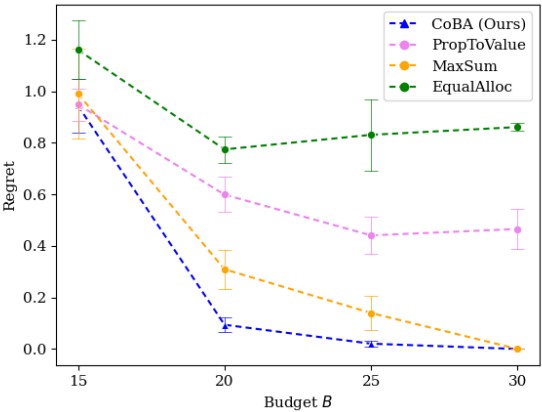

Figure 2: Experiment 2 results (Section 5.3).

Figure 1 provides the results of this Experiment. It plots the value of regret (normalized to $[0, 1]$ since different settings have different ranges for regret) as budget $B$ increases. We see that our algorithm performs better than all baselines. Our algorithm also retains its relatively lower regret at all values of $B$, providing empirical evidence of overall better regret performance.

### 5.3 Experiment 2 (randomized parameters)

To ensure that the results are not biased due to our choice of the representative set in Experiment 1, this experiment studies the performance of our algorithm when we *directly randomize the parameters* of the CPDs in each run, subject to realistic constraints. Specifically, in each run, we (1) randomly pick an $i \in \{1, ..., \lfloor |\mathsf{val}(C_1)|/2 \rfloor\}$, (2) distribute 20% of the probability mass randomly over the smallest $i$ values of $C_1$, and (3) distribute the remaining 80% of the mass over the remaining values of $C_1$. The smallest $i$ values of $C_1$ have higher value (i.e., the agent obtains higher rewards when the optimal action is chosen) than the other $C_1$ values. Intuitively, this captures the commonly observed 80-20 pattern (for example, 20% of the customers often contribute to around 80% of the revenue); but we randomize the other aspects. The full details of the parameterizations are given as part of the supplementary material (refer Appendix G). Averaging over runs provides an estimate of the performance of the algorithms on expectation. The number of samples in the initial dataset $\mathcal{D}_L$ is kept at $0.25 \cdot |\mathsf{val}(\mathcal{C}^A)| \cdot |\mathsf{val}(X)|$. The results are averaged over 50 independent runs; error bars display $\pm 2$ standard errors.

Figure 2 shows that our algorithm performs better than all baselines in this experiment. Our algorithm also demonstrates overall better regret performance by achieving the lowest regret for every choice of $B$.

### 5.4 Experiment 3 (calibrated using real-world data)

While Experiments 1 and 2 study purely synthetic settings, this experiment seeks to study the performance of our algorithm in *realistic scenarios*. We use the same causal graph used in the real world-inspired experiment in Section 4.2 of Subramanian & Ravindran (2022) and calibrate the CPDs using the data provided there. The graph is shown in Figure 3a; $\mathcal{C}^A = \{C_1, C_2\}$ and $\mathcal{C}^B = \{C_0\}$. For parameterizations, refer Appendix G.

The objective is to learn a policy that can assist salespeople by learning to decide how many outgoing calls to make in an ongoing deal, given just the type of deal and size of customer, so as to maximize a reward metric. The variables are related to each other causally as per the causal graph. The number of samples in the initial dataset $\mathcal{D}_L$ is kept at $0.125 \cdot |\mathsf{val}(\mathcal{C}^A)| \cdot |\mathsf{val}(X)|$. The results are averaged over 50 independent runs; error bars display $\pm 2$ standard errors. Figure 3 shows the results of the experiment. Our algorithm performs better than all other algorithms in this real-world inspired setting as well. Further, it retains its better performance at every value of $B$.

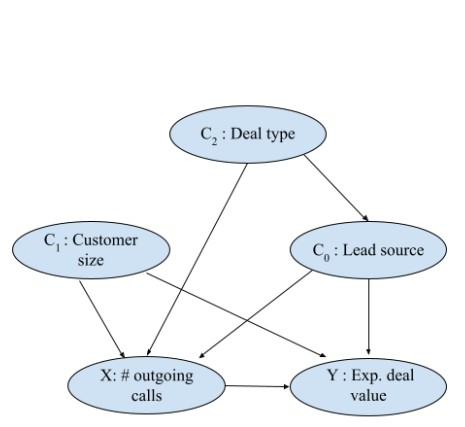

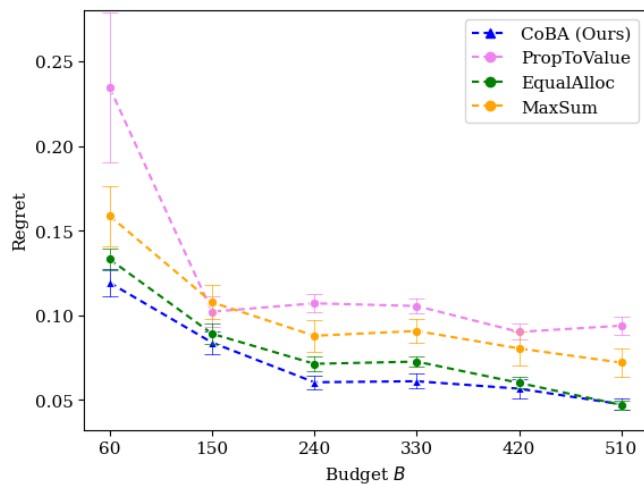

(a) Causal graph used in Experiment 3; taken from Subramanian & Ravindran (2022)

(b) Experiment results for real-world data inspired experiment (Experiment 3).

Figure 3: Causal graph and results of Experiment 3 (Section 5.4).

## 5.5 Experiments 4 through 7

Experiments 4 through 7 study *sensitivity of the results to key aspects* that define our settings. To aid this analysis, instead of regret, we consider a more aggregate measure which we call AUC. For any run, AUC is computed for a given algorithm by summing over $B$ the regrets for that algorithm; this provides an approximation of the area under the curve (hence the name). We then study the sensitivity of AUC to various aspects of the setting or environment.

**Experiment 4 ("narrowness" of $\mathcal{M}$)** We use the term "narrowness" informally. Since our algorithm CoBA exploits the information leakage in the causal graph, we expect it to achieve better performance when there is more leakage. To see this, suppose we do a forward sampling (Koller & Friedman, 2009) of $\mathcal{M}$; then, intuitively, more leakage occurs when more samples require sampling overlapping CPDs. For this experiment, we proxy this by varying $|\mathsf{val}(C_0)|$ while keeping $|\mathsf{val}(C_1)|$ fixed. The rest of the setting is the same as in Experiment 2. A lower $|\mathsf{val}(C_0)|$ means that the causal model is more "squeezed" and there is likely more information leakage. The results are averaged over 50 independent runs; error bars display $\pm 2$ standard errors. Figure 4 shows the results of this experiment. We see that our algorithm's performance remains similar (within each other's the confidence interval) for $|\mathsf{val}(C_0)|/|\mathsf{val}(C_1)| \in \{0.25, 0.375\}$, but significantly worsens when $|\mathsf{val}(C_0)|/|\mathsf{val}(C_1)| = 0.5$. However, our algorithm continues to perform better than all baselines for all values of $|\mathsf{val}(C_0)|/|\mathsf{val}(C_1)|$. Figure 4 broken down by $B$ is given in Appendix L.1.

**Experiment 5 (size of initial dataset)** The number of samples in the initial dataset $\mathcal{D}_L$ would impact the algorithm's resulting policy, for any given $B$. Specifically, we would expect that as the cardinality of $\mathcal{D}_L$ increases, regret reduces. For this experiment, we consider a uniformly exploring logging policy, and vary $|\mathcal{D}_L|$ by setting it to be $k \cdot |\mathsf{val}(\mathcal{C}^A)| \cdot |\mathsf{val}(X)|$, where $k \in \{0, 0.25, 0.5\}$. The rest of the setting is the same as in Experiment 2. The results are averaged over 50 independent runs; error bars display $\pm 2$ standard errors. The results are shown in Figure 5. We would expect the performance of all algorithms improve with increase in $k$ since that would give the agent better starting beliefs; this, indeed, is what we observe. Importantly, our algorithm performs better than all baselines in all these settings. Figure 5 broken down by $B$ is provided in Appendix L.2.

**Experiment 6 (choice of $\beta$)** Though we allow the cost function to be arbitrary, this experiment studies our algorithm's performance under two natural choices of $\beta(.)$ to test its robustness: (1) a constant cost function; that is, $\beta(x, \mathbf{c}^A, N_{x,\mathbf{c}^A}) \propto N_{x,\mathbf{c}^A}$, and (2) cost function that is inversely proportional to the likelihood of

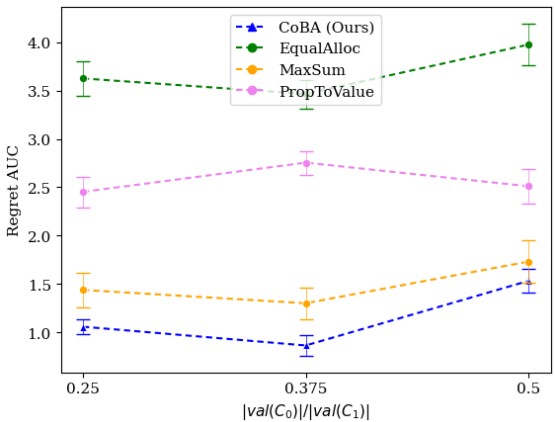
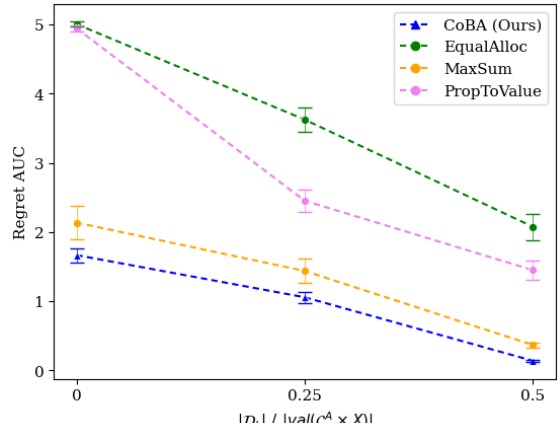

Figure 4: Experiment 4 results.                    Figure 5: Experiment 5 results.

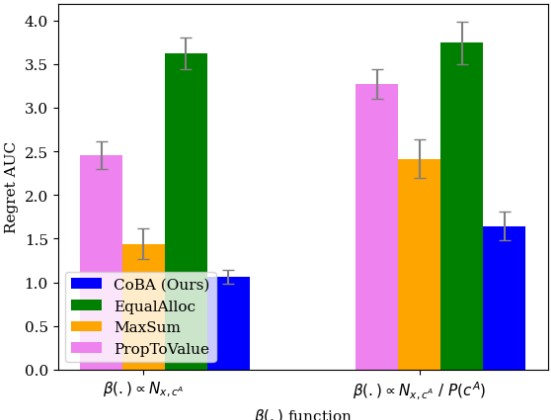

Figure 6: Experiment 6 results.

observing the context naturally (i.e., rarer samples are costlier); that is, $\beta(x, \mathbf{c}^A, N_{x,\mathbf{c}^A}) \propto \frac{N_{x,\mathbf{c}^A}}{\mathbb{P}[\mathbf{c}^A]}$. The rest of the setting is the same as in Experiment 2. The results are averaged over 50 independent runs; error bars display $\pm 2$ standard errors. Figure 6 shows the results. As expected, the choice of cost function does affect performance of all algorithms. However, our algorithm performs better than all algorithms for both cost function choices.

**Experiment 7 (misspecification of $\mathcal{G}$)** In real-world applications, the true underlying causal graph may not always be known. In this experiment, we study the impact of misspecification of $\mathcal{G}$ on the performance of our algorithm. Note that the formalism described in Section 3 does not necessitate that the agent knows the true underlying causal graph, but rather only that it knows a causal graph such that $\mathbb{P}$ factorizes according to it. This means that the graph $\mathcal{G}$ that the agent knows might include additional arrows not present in the true underlying graph. Intuitively, using such an imperfect graph would result in worsened performance by our algorithm since there are less overlapping information pathways to exploit. In Experiment 7a, we study this effect empirically by comparing results of Experiment 2 to the same experiment but with the causal graph having an extra edge: $C_1 \to Y$. The results are averaged over 25 independent runs; error bars display $\pm 2$ standard errors.

Figure 7 shows the results of the experiment. As expected, performance of our algorithm degrades when there is imperfect knowlege of the true underlying graph. However, our algorithm continues to perform better than all baselines, while also maintaining a similar difference in regret AUC compared to the baselines. Figure 7 broken down by $B$ is provided in Appendix L.3.

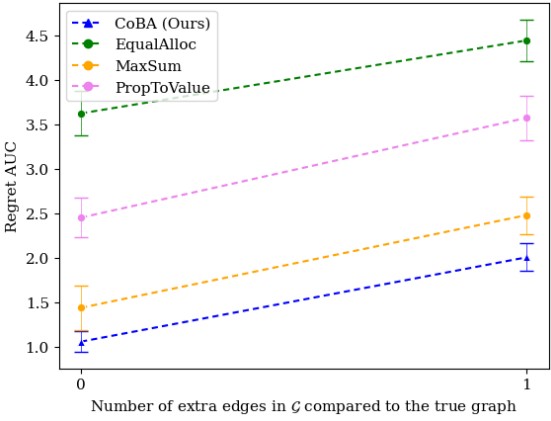

Figure 7: Experiment 7a results.

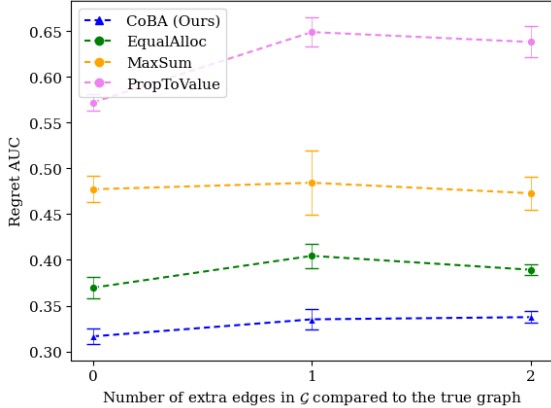

Figure 8: Experiment 7b results.

In Experiment 7b, we perform a similar analysis to the real-world inspired experiment presented in Section 5.4. Specifically, we compare the case where the true causal graph is known with the cases where there is a misspecification of the causal relationships between the context variables. To capture this, we add one edge $(C_1 \rightarrow C_0)$ to $\mathcal{G}$ and then one more edge $(C_2 \rightarrow C_1)$; we compare the performance under these two settings to the case where the true causal graph is known. The results are averaged over 25 independent runs; error bars display $\pm 2$ standard errors. The results in Figure 8 shows the results. In this case, the deterioration in performance due to misspecification of $\mathcal{G}$ is quite small for all algorithms. However, our algorithm continues to perform better than all baselines; it performs the best when the true graph is known.

## 6 Fairness

Fairness is becoming an increasingly important angle to discuss when designing machine learning algorithms. A common way to approach fairness is to ensure some subset of variables (assumed given to the algorithm), called "sensitive variables", is not discriminated against. Specific formal definitions of this discrimination give rise to different notions of fairness in literature (Grgić-Hlača et al., 2016; Dwork et al., 2012; Kusner et al., 2017; Zuo et al., 2022; Castelnovo et al., 2022).

**Counterfactual fairness** Counterfactual fairness is a commonly used notion of individual fairness. Intuitively, a *counterfactually fair* mapping from contexts to actions ensures that the actions mapped to an individual[8] are the same in a counterfactual world where a subset of sensitive contexts is changed. In our case, counterfactual fairness can be achieved by setting $\mathcal{C}^B$ to contain all the sensitive attributes. We provide a proof for this in Appendix I.

**Demographic parity** A common criterion for group-level fairness is Demographic Parity (Kusner et al., 2017). Our algorithm does not achieve demographic parity. However, in Appendix J, we suggest a way by which it can be achieved with some compromise to the agent's performance.

## 7 Conclusion and future research directions

This paper proposed a new contextual bandit problem formalism where the agent, which has access to qualitative causal side information, can also actively obtain a table of data in one shot, but at a cost and budget. We proposed a novel algorithm based on a new measure similar to entropy, and showed extensive empirical analysis of our algorithm's performance. We also showed theoretical results on soundness and regret. Furthermore, we studied the fairness implications of our algorithm. Possible directions of future research include allowing unobserved confounders and designing algorithms that meet population-level fairness criteria with minimial impact on performance.

---

[8]An individual is given by a specific choice of values for the context variables.

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

## A   Appendix: definition of Ent$^{new}$

This section is taken from Subramanian & Ravindran (2022). As discussed in Section 4.2, $\mathbb{P}(V|\mathbf{pa}_V) = \mathsf{Cat}(V; b_1, ..., b_r)$, where $(b_1, ..., b_r) \sim \mathsf{Dir}(\theta_{V|\mathbf{pa}_V})$. Here $\theta_{V|\mathbf{pa}_V}$ is a vector of length, say, $r$. Let $\theta_{V|\mathbf{pa}_V}[i]$ denote the $i$'th entry of $\theta_{V|\mathbf{pa}_V}$. We define an object called $\mathsf{Ent}$ that captures a measure of our knowledge of the CPD:

$$\mathsf{Ent}(\mathbb{P}(V|\mathbf{pa}_V)) \triangleq -\sum_i \left[ \frac{\theta_{V|\mathbf{pa}_V}[i]}{\sum_j \theta_{V|\mathbf{pa}_V}[j]} \ln\left( \frac{\theta_{V|\mathbf{pa}_V}[i]}{\sum_j \theta_{V|\mathbf{pa}_V}[j]} \right) \right]$$

We then define

$$\mathsf{Ent}^{new}(\mathbb{P}(V|\mathbf{pa}_V)) \triangleq \frac{1}{r} \sum_i \mathsf{Ent}(\mathsf{Cat}(b'_1, ..., b'_r))$$

where $(b'_1, ..., b'_r) \sim \mathsf{Dir}\left(..., \theta_{V|\mathbf{pa}_V}[i-1],\ \theta_{V|\mathbf{pa}_V}[i]+1,\ \theta_{V|\mathbf{pa}_V}[i+1], ...\right)$.

## B   Appendix: proof of soundness

We would like to show that $Regret \to 0$ as $B \to \infty$. As $B \to \infty$, in the limit, the problem becomes unconstrained minimization of $\Upsilon(\mathbf{N})$. Note that for all $\mathbf{N}$, $\Upsilon(\mathbf{N}) \geq 0$. Therefore, the smallest possible value of $\Upsilon(\mathbf{N})$ is 0.

First, note that $N_{x,\mathbf{c}^A} \to \infty, \forall (x, \mathbf{c}^A) \implies \Upsilon(\mathbf{N}) \to 0$. This is because $\forall (x, \mathbf{c}^A)$,

$$N_{x,\mathbf{c}^A} \to \infty \implies \frac{1}{1 + \ln(N_{x,\mathbf{c}^A} + 1)} \to 0$$

which, in turn, makes $Q(\mathbb{P}[V|\mathbf{pa}_V], \mathbf{N}) \to 0, \forall (V, \mathbf{pa}_V)$. From Equation 3, it is easy to see that this causes $\Upsilon(\mathbf{N}) \to 0$.

Also note that $\Upsilon(\mathbf{N}) \to 0 \implies N_{x,\mathbf{c}^A} \to \infty, \forall (x, \mathbf{c}^A)$. To see this, let $\Upsilon(\mathbf{N}) \to 0$, and consider the case where there exists a $(x, \mathbf{c}^A)$ such that $N_{x,\mathbf{c}^A}$ is finite. That means that there is at least one term of the form

$$\frac{1}{1 + \ln(N_{x',\mathbf{c}^{A'}} + 1)}$$

which occurs in the $Q(.)$ function of at least one CPD $\mathbb{P}[V|\mathbf{pa}_V]$, causing $Q(\mathbb{P}[V|\mathbf{pa}_V], \mathbf{N}) > 0$ since $\mathsf{Ent}^{new} > 0$. This, in turn, causes $\Upsilon(\mathbf{N}) \nrightarrow 0$, resulting in a contradiction.[9]

Thus, $N_{x,\mathbf{c}^A} \to \infty, \forall (x, \mathbf{c}^A) \iff \Upsilon(\mathbf{N}) \to 0$. In other words, each $(x, \mathbf{c}^A)$ gets a number of samples tending towards infinity if and only if $\Upsilon$ tends to 0. Thus, since $\Upsilon(\mathbf{N}) \geq 0$, Algorithm 1a will allocate $N_{x,\mathbf{c}^A} \to \infty, \forall (x, \mathbf{c}^A)$. Each CPD has at least one $(x, \mathbf{c}^A)$ whose samples will be used to update its beliefs in Algorithm 1a.[10] This means that each CPD will have its beliefs updated a number of times approaching infinity. Thus, for any $(V, \mathbf{pa}_V)$, $\hat{\mathbb{P}}[V|\mathbf{pa}_V] \to \mathbb{P}[V|\mathbf{pa}_V]$. As a result, we have that $\hat{\mathbb{P}} \to \mathbb{P}$.

Since the agent's policy constructed using $\mathbb{P}$ will necessarily be optimal, we have that $Regret \to 0$. This completes the proof.

## C   Appendix: proof of regret bound

In this section, we prove Theorem 4.2. The proof follows closely the one in Subramanian & Ravindran (2022) and adapts it to our setting.

First, we define the assumptions (A2) under which the theorem holds:

---

[9]This makes an implicit technical assumption that $\mathbb{P}[\mathbf{c}] > 0, \forall \mathbf{c}$ and that $\min(\mathsf{val}(Y)) > 0$. These are stronger assumptions than necessary, and could be weakened in the future.

[10]Due to the technical assumption in footnote 9

1. There is some non-empty past logged data ($|\mathcal{D}_L| > 0$), and it was generated by an (unknown) policy $\pi$ where every action has a non-zero probability of being chosen ($\pi(x|\mathbf{c}^A) > 0, \forall x, \mathbf{c}^A$). The latter is a commonly made assumption, for example, in inverse-propensity weighting based methods.

2. $|\mathcal{D}_L| \geq \alpha B$, for some constant $\alpha > 0$[11]. This is generally achievable in real world settings since we usually have fairly large logged datasets (or it is quite cheap to acquire logged data; for example, think of search logs), and for the bound to hold we technically can have a very small $\alpha$ as long as it is greater than 0.

3. The cost function $\beta$ is constant.[12] This is a common case in real world applications, especially when we do not have estimates of cost; in those cases, we typically assign a fixed cost to all targeted experiments.

## C.1 Expression for overall bound

First, note that Equation 3 in Subramanian & Ravindran (2022) remains the same even for our case. This is because it depends only on the factorization of $\mathbb{E}[Y|do(x), \mathbf{c}^A]$ (see Equation 1 in the main paper) and on the fact that in the evaluation phase the agent uses expected parameters of the CPDs (derived from its learned beliefs) to return an action for a given context.

Therefore, suppose, with probability $\geq 1 - \delta_{X, \mathbf{pa}_Y}$,

$$|\forall x, \hat{\mathbb{P}}(Y = 1|X, \mathbf{pa}_Y) - \mathbb{P}(Y = 1|X, \mathbf{pa}_Y)| \leq \epsilon_{X, \mathbf{pa}_Y}$$

and with probability $\geq 1 - \delta_{C|\mathbf{pa}_C}$,

$$\forall c, \ |\hat{\mathbb{P}}(C = c|\mathbf{pa}_C) - \mathbb{P}(C = c|\mathbf{pa}_C)| \leq \epsilon_{C|\mathbf{pa}_C}$$

where the expressions for $\delta_{X, \mathbf{pa}_Y}$, $\delta_{C|\mathbf{pa}_C}$, $\epsilon_{X, \mathbf{pa}_Y}$ and $\epsilon_{C|\mathbf{pa}_C}$ will be derived later in this section.

Then with probability $\geq 1 - \sum_{\mathbf{pa}_Y} \delta_{X, \mathbf{pa}_Y} - \sum_{C \in \mathcal{C}} \sum_{\mathbf{pa}_C} \delta_{C|\mathbf{pa}_C}$, for any given $\mathbf{c}^A$,

$$\text{Regret}(\mathbf{c}^A) = \mathbb{E}[Y|do(a^*), \mathbf{c}^A] - \mathbb{E}[Y|do(a_{alg}), \mathbf{c}^A] \leq 2\epsilon'_X + 3 \sum_{C \in \mathcal{C}^B} \epsilon'_C \tag{4}$$

where we define

$$\epsilon'_X \triangleq \sum_{\mathbf{pa}_Y} \mathbb{P}(\mathbf{pa}_Y|\mathbf{c}^A)\epsilon_{X, \mathbf{pa}_Y}$$

and

$$\epsilon'_C \triangleq \sum_{\mathbf{pa}_C} \mathbb{P}(\mathbf{pa}_C|\mathbf{c}^A)\epsilon_{C|\mathbf{pa}_C}$$

## C.2 Expressions for $\delta_{C|\mathbf{pa}_C}$ and $\epsilon_{C|\mathbf{pa}_C}$

Denote $M_{\mathcal{V}} \triangleq |\text{val}(\mathcal{V})|$. Let $L_{\mathbf{pa}_C}$ be the number of samples in $\mathcal{D}_L$ where $PA_C = \mathbf{pa}_C$. Now, our starting estimate of $\hat{\mathbb{P}}(C = 1|\mathbf{pa}_C)$ using $\mathcal{D}_L$ is computed as $(\theta^{(1)}_{C|\mathbf{pa}_C} + 1)/(L_{\mathbf{pa}_C} + 2)$. Since $\mathcal{D}_L$ is built by observing $C^A$ according to the natural distribution and choosing $X$ according to some (unknown) policy, the proof of Lemma A.1 in Subramanian & Ravindran (2022) can be followed if we replace $T'$ by $\alpha B$ since $|\mathcal{D}_L| \geq \alpha B$.

Therefore, suppose, with probability at least $1 - \delta^L_{C|PA_C}$, it is true that

$$\forall \mathbf{pa}_C, \ L_{\mathbf{pa}_C} \geq \alpha B \mathbb{P}(\mathbf{pa}_C, \mathbf{c}^A) - \epsilon^L_{PA_C}$$

---

[11]We also assume that $B$ is finite, as discussed earlier.
[12]Without loss of generality, we let this constant be equal to 1.

If the above event is true, then it is also true that with probability at least $1 - \delta_{C|\mathbf{pa}_C}$, it is true that

$$\forall c, \ |\hat{\mathbb{P}}(c|\mathbf{pa}_C) - \mathbb{P}(c|\mathbf{pa}_C)| \leq \sqrt{\left[\frac{2}{\alpha B \mathbb{P}(\mathbf{pa}_C, \mathbf{c}^A) - \epsilon^L_{C|PA_C}}\right] \ln\left(\frac{2}{\delta_{C|\mathbf{pa}_C}}\right)}$$

where

$$\epsilon^L_{C|PA_C} = \sqrt{\left[\frac{\alpha B}{2}\right] \ln\left(\frac{M_{PA_C}}{\delta^L_{C|PA_C}}\right)}, \quad M_{PA_C} = \prod_{C \in PA_C} M_C$$

Therefore, we have that

$$\epsilon_{C|\mathbf{pa}_C} = \sqrt{\left[\frac{2}{\alpha B \mathbb{P}(\mathbf{pa}_C, \mathbf{c}^A) - \epsilon^L_{C|PA_C}}\right] \ln\left(\frac{2}{\delta_{C|\mathbf{pa}_C}}\right)} \tag{5}$$

## C.3   Expressions for $\delta_{X, \mathbf{pa}_Y}$ and $\epsilon_{X, \mathbf{pa}_Y}$

Let $L_{x, \mathbf{pa}_Y}$ be the number of samples in $\mathcal{D}_L$ where $(X, PA_Y) = (x, \mathbf{pa}_Y)$. As before, recollect that our estimate of $\hat{\mathbb{P}}(Y = 1|x, \mathbf{pa}_Y)$ is computed as $(\theta^{(1)}_{Y|x, \mathbf{pa}_Y} + 1)/(L_{x, \mathbf{pa}_Y} + 2)$. Further, the mean of $L_{x, \mathbf{pa}_Y}$ is *at least* $|\mathcal{D}_L| \cdot \mathbb{P}(\mathbf{pa}_Y, \mathbf{c}^A) \cdot m > \alpha B m \mathbb{P}(\mathbf{pa}_Y, \mathbf{c}^A)$, where $m = \min_{x, \mathbf{c}^A} \pi(x|\mathbf{c}^A)$ and $\pi$ is the unknown logging policy. From our set of assumptions (A2), we have that $\pi(x|\mathbf{c}^A) > 0, \forall x, \mathbf{c}^A$; therefore, $m > 0$.

Given this, the proof of Lemma A.2 in Subramanian & Ravindran (2022) can be followed.

Therefore, suppose, with probability at least $1 - \delta^L_{X, PA_Y}$, it is true that

$$\forall (x, \mathbf{pa}_Y), \ L_{x, \mathbf{pa}_Y} \geq \alpha B m \mathbb{P}(\mathbf{pa}_Y, \mathbf{c}^A) - \epsilon^L_{X, PA_Y}$$

where

$$\epsilon^L_{X, PA_Y} = \sqrt{\left[\frac{\alpha B}{2}\right] \ln\left(\frac{M_X M_{PA_Y}}{\delta^L_{X, PA_Y}}\right)}$$

If the above event is true, then it is also true that with probability at least $1 - \delta_{X, \mathbf{pa}_Y}$, it is true that

$$\forall x, |\hat{\mathbb{P}}(Y = 1|x, \mathbf{pa}_Y) - \mathbb{P}(Y = 1|x, \mathbf{pa}_Y)| \leq \sqrt{\left[\frac{2}{\alpha B m \mathbb{P}(\mathbf{pa}_Y, \mathbf{c}^A) - \epsilon^L_{X, PA_Y}}\right] \ln\left(\frac{2 M_X}{\delta_{X, \mathbf{pa}_Y}}\right)}$$

Therefore, we have that

$$\epsilon_{X, \mathbf{pa}_Y} = \sqrt{\left[\frac{2}{\alpha B m \mathbb{P}(\mathbf{pa}_Y, \mathbf{c}^A) - \epsilon^L_{X, PA_Y}}\right] \ln\left(\frac{2 M_X}{\delta_{X, \mathbf{pa}_Y}}\right)} \tag{6}$$

## C.4   Final bound

Now, we can plug the Equations 5 and 6 back into Equation 4, and following the same union bound trick as in Subramanian & Ravindran (2022) and some algebra, we get that for any $0 < \delta < 1$, with probability $\geq 1 - \delta$,

$$Regret \leq 3\mathbb{E}_{\mathbf{pa}_Y, \mathbf{c}^A} \left( \sqrt{ \left[ \frac{2}{\alpha m B \mathbb{P}(\mathbf{pa}_Y, \mathbf{c}^A) - \epsilon^L_{X, PA_Y}} \right] \ln \left( \frac{2 M_X (M_\mathcal{C} + |\mathcal{C}|)}{\delta} \right) } \right)$$

$$+ 3 \sum_{C \in \mathcal{C}^B} \mathbb{E}_{\mathbf{pa}_C, \mathbf{c}^A} \left( \sqrt{ \left[ \frac{2}{\alpha B \mathbb{P}(\mathbf{pa}_C, \mathbf{c}^A) - \epsilon^L_{PA_C}} \right] \ln \left( \frac{2 (M_\mathcal{C} + |\mathcal{C}|)}{\delta} \right) } \right) \quad (7)$$

where

$$\epsilon^L_{PA_C} = \sqrt{ \left[ \frac{\alpha B}{2} \right] \ln \left( \frac{M_{PA_C} (M_\mathcal{C} + |\mathcal{C}|)}{\delta} \right) }, \quad \epsilon^L_{X, PA_Y} = \sqrt{ \left[ \frac{\alpha B}{2} \right] \ln \left( \frac{M_X M_{PA_Y} (M_\mathcal{C} + |\mathcal{C}|)}{\delta} \right) }$$

It can be simplified as presented in Theorem 4.2 as:

$$Regret \in O \left( |\mathcal{C}| \sqrt{ \left( \frac{1}{mB - \epsilon} \right) \ln \frac{M_X M_\mathcal{C}}{\delta} } \right)$$

where

$$\epsilon \in O \left( \sqrt{ B \ln \frac{M_X M_{PA_Y} M_\mathcal{C}}{\delta} } \right)$$

This completes the proof.

## D  Regret behavior for large values of $B$

Figures 1 and 2 provided regret behavior for small values of $B$. We are primarily interested in small-budget behavior since that occurs more commonly in practice; for example, budgets exclusively for experimentation in software teams in often quite low.

However, it is also interesting to look at regret behavior as $B$ becomes large. Specifically, we increase $B$ large enough that all algorithms converge to optimal (or very close to optimal). We do this for Experiments 1 and 2. Figures 9 and Figures 10 provide the results. Note that the Figures 1 and 2 just zoom into these plots for small $B$ (i.e., $B$ between 15 and 30).

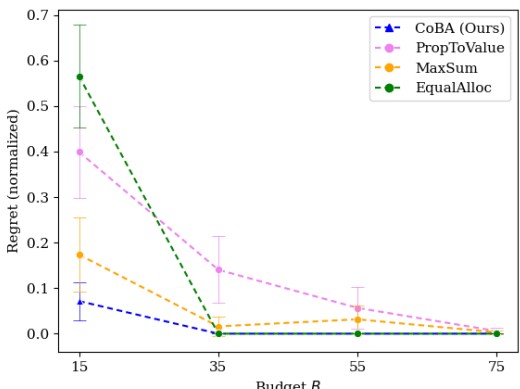

Figure 9: Experiment 1 results (Section 5.2) for large values of $B$.

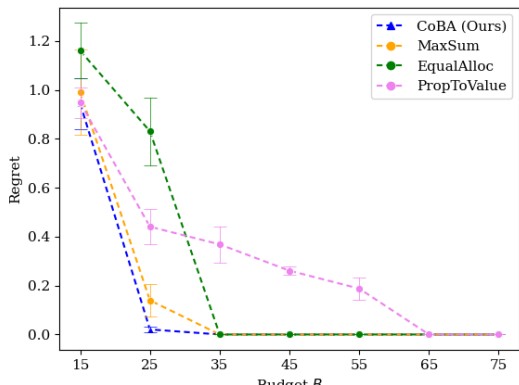

Figure 10: Experiment 2 results (Section 5.3) for large values of $B$.

PropToValue is the slowest to converge to the optimal policy in both instances, though it demonstrates better low-budget behavior than EqualAlloc. MaxSum has the best low budget behavior among the baselines because it maximizes the total number of samples within that low budget; it, as $B$ gets larger, EqualAlloc catches up (and even outperforms it) as it explores the context-action space better. In both experiments, however, our algorithm converges to an optimal policy faster than all baselines.

## E    Appendix: Understanding the reason for better performance of our algorithm

As discussed in Section 4, our algorithm balances the trade off between allocating more samples to context-action pairs that are higher value according to its beliefs and allocating more samples for exploration, while taking into account information leakage due to the causal graph. To understand this in more detail, we consider Setting 1 of Experiment 1, and zoom into the case where $B = 20$. We do 50 independent runs and plot the frequency of choosing samples containing different value of $C_1$. We show this for our algorithm and all baselines (except EqualAlloc since it is obvious how it allocates).

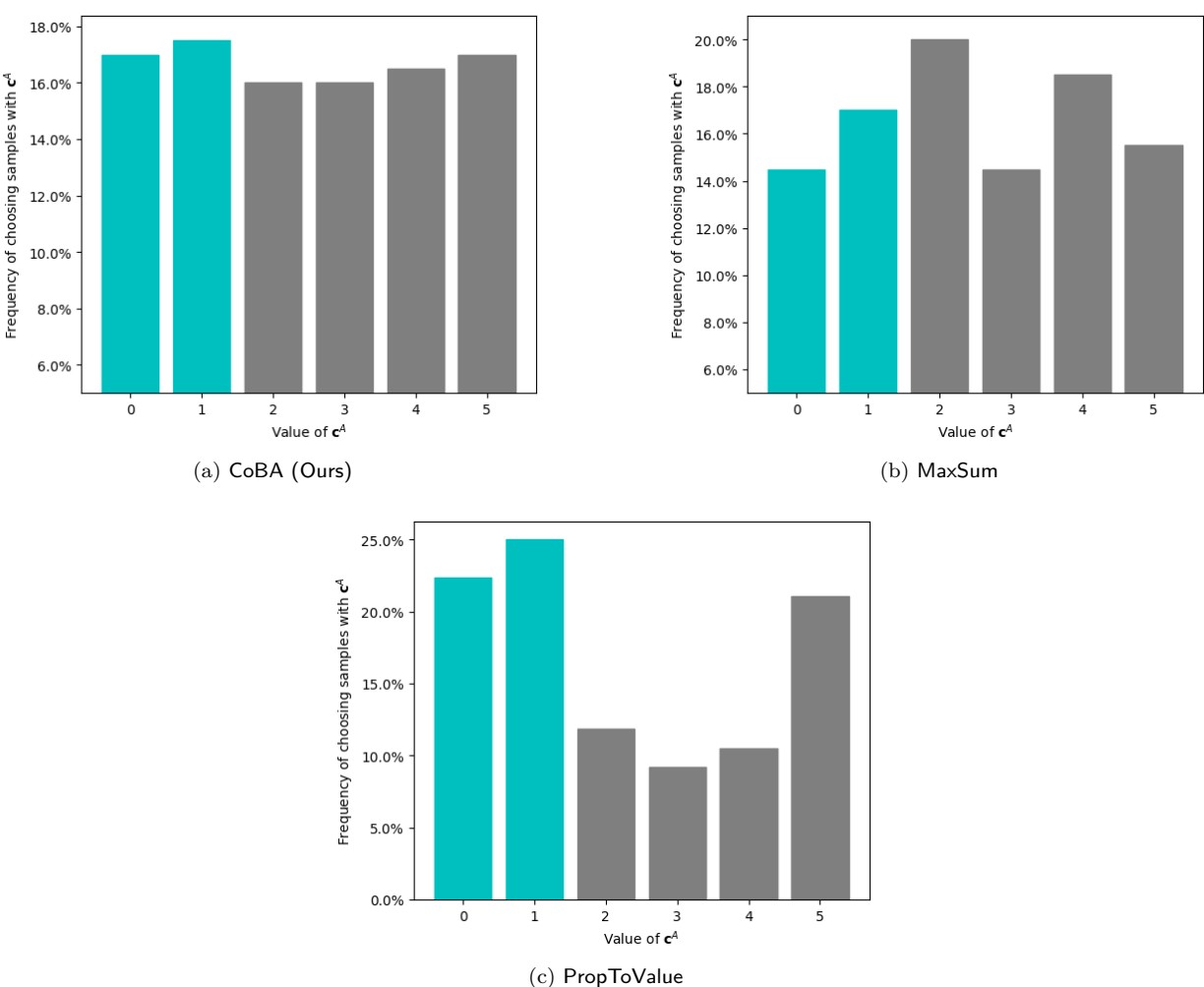

Figure 11: Frequency of choosing or encountering each value of $\mathcal{C}^A$. Highlighted in teal color are the 'high-value' contexts (i.e., contexts for which learning the right actions provides higher expected rewards).

Figure 11 shows the results of this experiment. MaxSum allocates lesser number of samples than our algorithm to the two context values ($C_1 \in \{0, 1\}$) that are high value. PropToValue over-allocates to these two context

values, resulting in poor exploration of other contexts. Our algorithm, in contrast, allocate relatively more to the high-value contexts, while also maintaining good exploration of other contexts.

## F  Appendix: Intuition for why EqualAlloc performs worse in Experiment 2

First, if we look at the figures in the main paper along with Figure 10, we see that that EqualAlloc performs worse only for low values of $B$, and significantly improves its performance once $B$ becomes 35.

To understand the intuition, first note that Experiment 2 captures the 80-20 rule (while randomizing other aspects). It is easy to see that EqualAlloc overallocates to the lower-value 80% of contexts. PropToValue, on the other hand, allocates more to the 20%, and therefore performs better for lower budgets (i.e., makes better use of the small budget). However, as budget increases, PropToValue fails to explore well, causing EqualAlloc to outperform it. MaxSum outperforms EqualAlloc in lower budgets because allocations which allocate more to the higher-value contexts are also possible solutions for MaxSum, and therefore on average it performs better than EqualAlloc which overallocates to the low-value contexts consistently. However, as $B$ becomes larger, EqualAlloc catches up, as seen in Figure 10.

## G  Appendix: Regarding parameterizations

For Experiments 1 through 3, we follow parameterizations very similar to the one used in Subramanian & Ravindran (2022). Please see `README.md` in the `Code` folder in `SupplementaryMaterial.zip` for the full parameterizations of all experiments.

## H  Appendix: How to implement our algorithm in practice

In our experiments, we have used the `scipy.optimize.differential_evolution` solver[13] from the `scipy` Python library to solve the problem in Section 4.2. This solver implements differential evolution,[14] which is an evolutionary algorithm which makes very few assumptions about the problem. The `scipy.optimize.differential_evolution` method is quite versatile; for example, it allows the specification of the objective function as a Python callable, and also allows arbitrary nonlinear constraints of type `scipy.optimize.NonlinearConstraint`.

However, a practitioner can use any suitable optimization algorithm or heuristic to solve the problem in Section 4.2.

## I  Appendix: Proof of counterfactual fairness

To prove counterfactual fairness, first note that the learned policy is a map $\hat{\phi} : \mathsf{val}(\mathcal{C}^A) \to \mathsf{val}(X)$; during inference, for any given $\mathbf{c}^A$, the value of $X$ is intervened to be set to $\hat{\phi}(\mathbf{c}^A)$. Following the notation in Pearl (2009a), we let $\hat{\phi}_{\mathcal{C}^B \leftarrow \mathbf{c}^{B\prime}}(\mathbf{c}^A)$ denote $\hat{\phi}(\mathbf{c}^A)$ in the counterfactual world where the variables in $\mathcal{C}^B$ are set equal to $\mathbf{c}^{B\prime}$. To achieve counterfactual fairness[15], we want that, for all $\mathbf{c}^A, \mathbf{c}^B, \mathbf{c}^{B\prime}, x$,

$$\mathbb{P}\left[\hat{\phi}_{\mathcal{C}^B \leftarrow \mathbf{c}^B}(\mathbf{c}^A) = x|\mathbf{c}^A, \mathbf{c}^B\right] = \mathbb{P}\left[\hat{\phi}_{\mathcal{C}^B \leftarrow \mathbf{c}^{B\prime}}(\mathbf{c}^A) = x|\mathbf{c}^A, \mathbf{c}^B\right]$$

Now, under the assumptions in Section 3.1, the conditional independences in $\mathcal{G}$ imply that we have

$$\mathbb{P}\left[\hat{\phi}(\mathbf{c}^A) = x|\mathbf{c}^A, \mathbf{c}^B\right] = \mathbb{P}\left[\hat{\phi}(\mathbf{c}^A) = x\right], \forall \mathbf{c}^B$$

This gives us that

$$\mathbb{P}\left[\hat{\phi}_{\mathcal{C}^B \leftarrow \mathbf{c}^{B\prime}}(\mathbf{c}^A) = x|\mathbf{c}^A, \mathbf{c}^B\right] = \mathbb{P}\left[\hat{\phi}(\mathbf{c}^A) = x\right], \forall \mathbf{c}^A, \mathbf{c}^B, \mathbf{c}^{B\prime}, x$$

which satisties the counterfactual fairness condition.

---

[13]https://docs.scipy.org/doc/scipy/reference/generated/scipy.optimize.differential_evolution.html
[14]https://en.wikipedia.org/wiki/Differential_evolution
[15]Our definition of counterfactual fairness draws from the definitions in Kusner et al. (2017); Zuo et al. (2022).

## J    Appendix: Demographic Parity

A common criterion for group-level fairness is Demographic Parity (Kusner et al., 2017). Demographic Parity (DP) requires that the distribution over actions remains the same irrespective of the value of the sensitive variables. Formally, it requires that

$$\mathbb{P}\left[\hat{\phi} = x | \mathbf{c}^B\right] = \mathbb{P}\left[\hat{\phi} = x | \mathbf{c}^{B\prime}\right], \ \forall \mathbf{c}^{B\prime}$$

Our algorithm, however, does *not* guarantee DP:

$$\mathbb{P}\left[\hat{\phi} = x | \mathbf{c}^B\right] = \sum_{\mathbf{c}^A} \mathbb{P}[\mathbf{c}^A | \mathbf{c}^B] \cdot \mathbb{P}[\hat{\phi}(\mathbf{c}^A) = x]$$

which may not equal $\mathbb{P}\left[\phi = x | \mathbf{c}^{B\prime}\right]$ since $\mathbb{P}[\mathbf{c}^A | \mathbf{c}^B]$ may not equal $\mathbb{P}[\mathbf{c}^A | \mathbf{c}^{B\prime}]$.

However, we discuss one way through which DP can be achieved, but with a reduction in agent's performance. Specifically, we can achieve DP by ensuring that the agent acts according to a fixed policy irrespective of the value of $\mathbf{c}^A$. Intuitively, we construct a fixed policy that maximizes rewards given the agent's learned beliefs.

Specifically, let $\hat{\psi}(x, \mathbf{c}^A) \triangleq \hat{\mathbb{E}}[Y | do(x), \mathbf{c}^A]$. Assume the fixed policy is probabilistic. Therefore, we're interested in a policy $q$ which is a distribution over $|\mathsf{val}(X)|$. Denoting $q^{(x)} \triangleq q(x)$, we solve the following optimization problem:

$$< ..., q^{(x)}, ... >= \underset{<...,q^{(x)\prime},...>}{\arg\max} \sum_x \left[ q^{(x)\prime} \left[ \sum_{\mathbf{c}^A} \hat{\mathbb{P}}[\mathbf{c}^A] \cdot \hat{\psi}(x, \mathbf{c}^A) \right] \right]$$

subject to

$$\sum_x q^{(x)\prime} = 1$$

It is easy to see that one global optimum to this involves assigning a probability of 1 to an action $x$ that results in the largest value of $\sum_{\mathbf{c}^A} \hat{\mathbb{P}}[\mathbf{c}^A] \cdot \hat{\psi}(x, \mathbf{c}^A)$. That is, choose $x$ such that

$$x = \underset{x'}{\arg\max} \sum_{\mathbf{c}^A} \hat{\mathbb{P}}[\mathbf{c}^A] \cdot \hat{\psi}(x', \mathbf{c}^A)$$

and let $q(x) = 1$, and $q(x') = 0, \forall x' \neq x$. Note that this fixed policy would perform worser on expectation than the context-specific policy learned by the agent in the main part of the paper. This, however, is a cost that can be paid to achieve DP.

## K    Appendix: Information leakage when $\mathcal{C}^B = \emptyset$

Our formalism (Section 3) allows $\mathcal{C}^B$ to be empty. In this case, there is still information leakage possible, which our algorithm can exploit. To see this, consider the same graph we used for Experiment 1 (Section 5), but let $\mathcal{C}^A = \{C_1, C_0\}$. Consider the two pairs $(x, c_0, c'_1)$ and $(x, c_0, c_1)$. It turns out that $\mathbb{P}[y | do(x), c_0, c'_1] = \mathbb{P}[y | do(x), c_0, c_1] = \mathbb{P}[y | x, c_0]$.

Thus, there is information leakage arising due to the conditional independencies arising from the causal graph, even when $\mathcal{C}^B = \emptyset$. Our algorithm exploits this structure through $Q(\mathbb{P}[Y | x, c_0], \mathbf{N})$.

## L    Appendix: Detailed plots

### L.1    Related to Experiment 4

Figure 12 shows the breakdown on Figure 4 for each value of $B$. For $B = 15$, there is no clear pattern; this is because the budget is so small that they all tend to learn poor policies. But when we increase $B$, we see

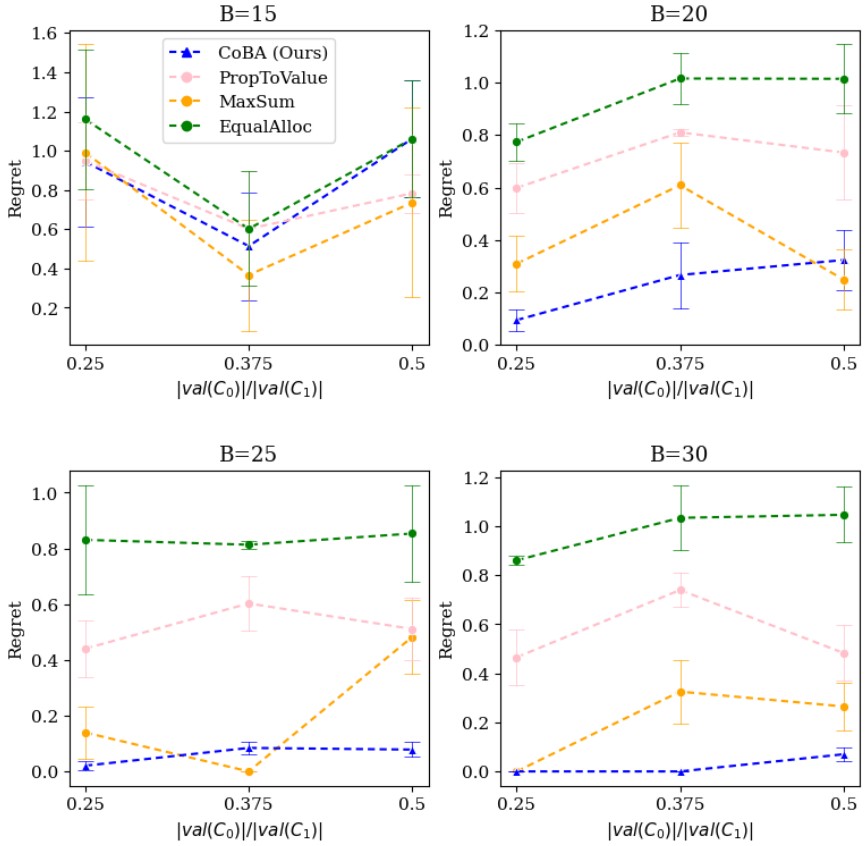

Figure 12: Detailed plots related to Experiment 4 (Section L.1).

that our algorithm performs better when the graph is the most squeezed – like we had discussed in the main part of the paper. Further, in the most squeezed setting (lowest value of x-axis), our algorithm's difference from the next best algorithm is larger for smaller $B$, and decreases as $B$ increases; this is because a larger budget reduces the advantage that our algorithm gains from better utilization of information leakage.

### L.2 Related to Experiment 5

Figure 13 shows the breakdown of Figure 5 for each value of $B$. We see that for each value of $B$, as $\frac{\mathcal{D}_L}{\mathsf{val}(\mathcal{C}^A \times X)}$ increases, regret decreases – similar to what happens in the aggregated plot. Further, as $B$ increases, $0$ regret is achieved faster – again in line with expectations.

### L.3 Related to Experiment 7

Figure 14 shows the breakdown of Figure 7 for each value of $B$. For $B = 15$, there is no clear pattern; this is because the budget is so small that they all tend to learn poor policies. But when we increase $B$, we see that all algorithms perform worser when $\mathcal{G}$ does not match the true underlying causal graph. Further, our algorithm continues retain its better performance over all baseline for all values of $B$.

## Broader Impact Statement

This work provides an improved mathematical framework and general-purpose algorithm for contextual bandits. We do not use any data that contains any personally identifiable information or sensitive personally identifiable information. To the best of our knowledge, we do not believe there were any ethical issues

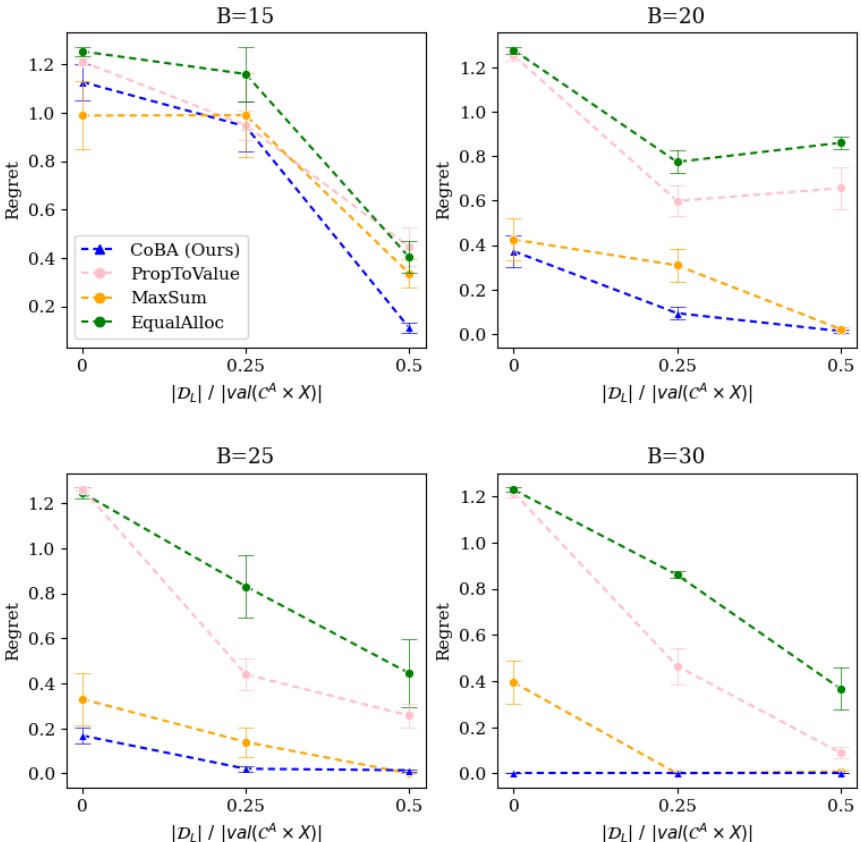

Figure 13: Detailed plots related to Experiment 5 (Section L.2).

associated with the development of this work. Further, given the nature of the work as foundational and introducing a new algorithm (and not specific to an application), we do not foresee any specific potential negative ethical issues created by this work. However, we do point out that researchers utilizing this method to their specific applications should adhere to ethical standards of their own (e.g., by avoiding targeting interventions on subpopulations based on racial attributes, or by ensuring that rewards are not defined in a way that incentivizes learning discriminatory behavior).

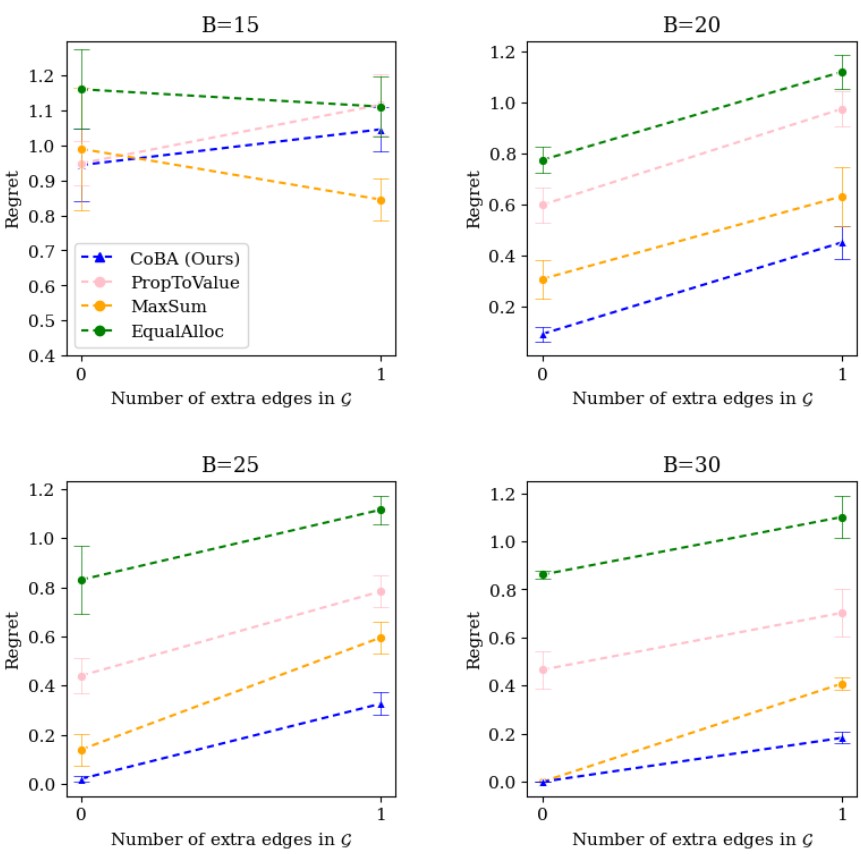

Figure 14: Detailed plots related to Experiment 7 (Section L.3).

