# OpenReview forum: "CoBA: Causal Contextual Bandits with Active Data Integration"
_TMLR — Rejected by TMLR_

### Review · Reviewer_4p2X · 2023-07-13

**Summary Of Contributions:**

## High-Level Overview

One-shot decision-making has a wide range of applications, from software experimentation, recommendation systems, marketing campaign design, etc.  In all of these settings, the objective is to learn a near-optimal policy that maps contexts (information about the current decision) to actions, with the hopes of achieving good downstream rewards.  In many of these settings, the algorithm has offline data available apriori and a budget to be able to select new context action pairs to observe their performance.  For example, consider the software design optimization setting. The agent has access to historical data on different features of the software and the relevant user metrics. However, subject to an experimentation budget, they can perform additional roll-outs of different feature combinations to test the metrics, prior to deciding the final software decisions.  At a high level, the main contributions for the authors are as follows. $(i)$ The authors study how to actively obtain multiple samples in contextual bandits with an additional causal graph modelling how the context variables interact.  $(ii)$ They propose a new algorithm based on minimizing an entropy measure for selecting the "one-shot" data points to experiment on. $(iii)$ They provide numerical studies showing the performance of their algorithm against other benchmarks.

## Main Model
The underlying problem is modeled as a causal model $M$ which depends on a DAG $G$ and a probability distribution $P$ which factorizes over it.  The set of variables in the graph contains the context variables (both main and auxiliary ones that the policy cannot depend on), the action, as well as the outcome reward.  The agent only knows the graph $G$ but not the probability distribution over it.

The agent has access to logged offline data collected from an unknown behavioural policy.  The data consists of samples $(c_i, x_i, y_i)$ where $c_i$ is context, $x_i$ is action, and $y_i$ is feedback.  There is a cost $\beta$ for obtaining a specific sample, and the goal of the algorithm is to pick $N_{(x,c)}$ for the number of samples to collect from an $(x,c)$ pair subject to a maximum budget of $B$.

The agent's goal is to learn a policy $\hat{\phi}$ mapping contexts $C$ to actions $X$ such that it minimizes regret compared to the optimal policy averaged over the context distribution from the underlying model.

## Assumptions

The authors make several simplifying assumptions in the work:
1. The set of context variables is known and finite
2. The graph has a single outgoing edge from action $X$ to reward $Y$
3. There are no unobserved confounders
4. If $C$ confounds $C'$ which is a main context variable then $C$ is also a main context variable.  This is needed for the fairness results and simplifies the algorithm design.

## Main Algorithm

First note that the expect reward can be written as an expectation taken with respect to the model of the causal graph.  Hence, given an estimate of the underlying model for the causal graph, a simple estimate for the optimal policy is to solve for the optimal policy based on the estimated model.  As such, in order to develop the learned policies the authors simply use a plug-in approach.  The main technical question is deciding how to select the samples $N_{(x,c)}$ for the different context action pairs subject to the budget constraints $B$.

To this end, the authors define an entropy-like measure $\Psi(N)$ using inspiration from Subramanian and Ravindran (2022).  Given this, the sampling approach optimizes this subject to the budget constraints.

## Theoretical Results

The authors show a single theoretical result for their algorithm, notably that as the budget $B$ approaches infinity then the agent's regret will tend towards zero.  However, this is true for any algorithm which in the limit collects infinite samples for every context pair, achieved by other benchmarking algorithms such as proportional or equal allocation.

## Experimental Results

The authors complement the theoretical results with experimental results highlighting the performance of their algorithms.  The results show that their approach (COBA) outperforms other heuristics including
1. Equal Allocation
2. Proportional Allocation
3. Maximizing the total number of samples obtained

## Fairness Results

Lastly, the authors provide a brief discussion on the fairness property of the algorithm.  If the "auxiliary" context variables contains all protected attributes, then they show that their resulting policy satisfies counterfactual fairness.



**Audience:**

Yes

**Broader Impact Concerns:**

As part of the Appendix the authors provide a sufficient broader impact statement highlighting that $(i)$ the work provides a general purpose algorithm outside of any particular application and $(ii)$ researchers utilizing the method should adhere to ethical standards for their specific context.

**Claims And Evidence:**

Yes

**Requested Changes:**

## Requested Changes
- Theoretical Results: The theoretical results provided only provide soundness, which is achieved by a wide class of algorithms. Since the paper is relatively theoretical, it would be interesting to see more discussion around potential rates.  For example, what is the regret rates provided by Subramanian and Ravindran (2022), and could those techniques be used similarly in this setting?
- Scalability: There is limited discussion on the scalability of the experiments and the assumptions.  The authors assume a finite context space (which doesn't capture most real-world settings) and this is heavily used in the experiment and algorithm design.  Moreover, the authors only comment that the optimization problem is solved approximately using scipy.  However, as written it is a combinatorial optimization problem, and so the authors should comment more on how to implement their algorithm in practice.
- Fairness: The fairness discussion seemed added on as an after-thought (including the separation of the context variables).
- Experiments: To help make the theoretical results more interesting, it would potentially be interesting to see the following experiments:
1) Scale the $B$ to look at asymptotic learning efficiency of your algorithm compared to the benchmarks
2) Implement the "optimal" allocation discussed in Remark on page 8

## Questions
- Is the simplifying assumption on the "hidden" context variables only required for the fairness model?
- Can you comment on the scalability and implementation details of the algorithm?
- What is the optimal solution to Section 4.2 without any logged data? That might help build some more intuition, because taking into account the causal model will return a solution which isn't equal allocation.
- Any ability to compare algorithm performance to Subramanian and Ravindran (2022)?
- Compare to optimal allocation for small scale experiments?
- Intuition in Experiment 2 that equal allocation is performing worse?

## Minor Comments
- Abstract is nicely written, clear and concise
- The introduction provides nice context in the model.  Maybe it would be helpful to emphasize how the one-shot comes in. Moreover, under the advertisement model initially, the one-shot feels like picking users (obviously tough), but because it is aggregate over contexts that is more feasible.
-  "of of" on top of page two
- Section 1.3 provides a great summary of the paper
- Table 1 is very helpful for the readability of the paper
- First paragraph under Section 5.2 is a bit tough to read
- "As" in mid-sentence under Appendix B

**Strengths And Weaknesses:**

## Strengths
- The paper is very well written and easy to follow. The table of notation is easy to refer back to during the technical discussions, and the introduction provides a succinct overview of the entire paper.
- The authors consider a very well-motivated model, and provide one of the first works to consider causal contextual bandits with the one-shot data collection setting.
- The empirical results help complement their algorithm design and analysis by showing the performance of their algorithm (COBA) compared to several other benchmarks.

## Weaknesses
- The only theoretical guarantees are soundness, which is very weak and achieved by a wide class of algorithms.
- The experimental results, to complement the soundness, only look at small-scale (limited budget) performance.  However, since soundness provides no regret rates it would be more interesting to look at asymptotics to show that their algorithm learns more efficiently than equal-allocation benchmarks.
- The fairness model considered is not very well fleshed out and seems added only as an afterthought.

---

### Review · Reviewer_mgV9 · 2023-07-25

**Summary Of Contributions:**

The paper proposes a method to select a set of bandit experiments to perform in a single decision instant. The authors propose a method to integrate the information provided by historical data and a causal graph to learn the optimal policy to perform in such a contextual bandit setting. A guarantee of convergence is provided, and several experiments and sensitivity analyses have been performed to test the capabilities of the approach.

**Audience:**

Yes

**Broader Impact Concerns:**

I do not see any direct impact on society or ethical impact. The method has been also characterized in terms of fairness.

**Claims And Evidence:**

Yes

**Requested Changes:**

1) Section 1.2: i suggest you to add some more details about the definition of fairness you are going to consider.
2) define the symbol do(x)
3) add formulas punctuation
4) a parenthesis is missing in the second to last line on page 4
5) Section 3.1 I think that an example of the causal graph would help in understanding the setting.
6) add a textual description of the proposed algorithm since the pseudocode is not self-explanatory
7) I see the value of Theorem 1, but I think that if you are in a bandit setting, you should somehow also try to quantify the convergence order of the regret (maybe only asymptotically). this would give a more robust justification for the work you proposed.
8) from the introduction, I understood that the method might have been instantiated even in the case no prior information is available. Instead, if I look at Alg. 1, it shows that historical data are necessary. Can you comment more on that?
9) What about comparing your proposed method with online learning strategies? for instance, the one from Subramanian 2022, to analyze how much we are losing because we cannot act in an online way.
10) Demographic parity: I am unsure why you opt to leave this part in the appendix.





**Strengths And Weaknesses:**

I think the setting is derivative, but the idea of a one-shot learning procedure is somehow interesting. However, I think that the theoretical guarantees provided are too not detailed enough to be relevant to the problem.

Moreover, I think that the setting is not motivated fully. I think that the real testing campaigns are run in batches. Therefore, this work should also take into account the case in which the learning procedure is repeated multiple times.

I am unsure about the title you provided for the paper. Indeed, your modeling is somehow not applicable to a sequential decision-making setting since the choice you provided is a single planning strategy.
I think that what you are proposing is similar to the work in the learning from logged bandit feedback, but not totally coherent with it. Therefore, my suggestion is to try to make the proposal of the paper more clear from the title.

---

### Review · Reviewer_et4X · 2023-07-25

**Summary Of Contributions:**

This paper considered causal contextual bandits with single batch data acquisition, a setting similar to active learning. Given an initial dataset to warm start, the bandit policy has one shot to query a batch of actions under budget constraints. The goal is to find the best policy that minimizes simple regret. The paper also considered the presence of a causal graph. The authors proposed CoBA algorithm that solves a nonlinear optimization problem with nonlinear and integer constraints where the objective considers an entropy-like measure to capture the impact of new samples. An asymptotic theoretical result is provided that as the budget goes to infinity, the simple regret goes to zero. Extensive experiments on synthetic and real-world data validated the effectiveness of the proposed algorithm.


**Audience:**

Yes

**Claims And Evidence:**

No

**Requested Changes:**

1. Please clarify the relation to active learning and batched bandits.

2. Please clarify if the results in the paper could be generalized to multi-armed bandits or contextual bandits.

3. Please consider to anlyze the algorithm theoretically with finite budgets. While study general cost functions would be ideal, simple cost function such as proportional to the number of samples could be an informative first step.


**Strengths And Weaknesses:**

Strengths:

1. The problem of one shot data acquisition in contextual bandits is a novel and well-motivated setting that could be interesting to the readers.

2. The proposed algorithm appears to be practical.

3. The empirical evaluations are well designed and thorough.

Weaknesses:

1. [Related work] The relation to existing works is not clearly explained in related work section. First, active learning is highly related to the setting. The authors mentioned that "the agent receives outcomes only for actions that were taken" which is indeed the setting of pool-based active learning. One difference I can think of is that the objective here is to minimize simple regret, but it needs further clarification. Second, the paper is also related to batched bandits where the time horizon $T$ could be (much) smaller than batch size. This paper could be viewed as an extreme case of batched bandits with $T=1$ and the authors should explain why existing results are not related (apart from the causal setting).

2. [Scope] The authors build the problem setting of single shot data acquisition based on causal contextual bandits [1]. The scope of the work might be limited since single shot data acquisition problem has not been studied in basic settings such as multi-armed bandits or contextual bandits yet (according to the related work section). The authors should clarify if the results in the paper could be generalized to multi-armed bandits or contextual bandits, which will generate greater impact.

3. [Theoretical result] The asymptotic theoretical result cannot justify the proposed algorithm: taking budget to infinity makes the optimization problem unconstrained and can only represent the case of training with infinite samples. This cannot justify the selective data acquisition problem considered in the paper. It is important to understand the trade-off between budgets and regrets even if the analysis is limited to simple cost function such as proportional to the number of samples (the one used in Experiment 6).

[1] Chandrasekar Subramanian and Balaraman Ravindran. Causal contextual bandits with targeted interventions. In Proceedings of the Tenth International Conference on Learning Representations, 2022.

---

### Decision · Action_Editors · 2023-09-18

**Recommendation:** Reject

**Comment:**

There was significant improvement after the revision; however, the reviewers still have concerns leaning towards rejecting the current version of the paper. The AE believes that the submission could be significantly improved by incorporating the comments provided below.

### Theoretical guarantee

The soundness result is not solid, as it can be achieved by other wide classes of algorithms. The authors have added new results on the regret bound in their revision; however, this result is still not strong enough, especially given strict assumptions such as the cost being equal to one. As a bandit paper, it should include a more substantial regret bound theorem. Additionally, the reviewers and AE strongly recommend including a discussion to help interpret or compare their bound with prior works.

### Motivation and positioning

The reviewers and AE are not fully convinced by the suggested problem setup. The authors could take into account Reviewer mgV9's comment and strengthen the motivation (Sec. 1.1), for example, by including a visual illustration of the practical scenario of the considered problem. Additionally, the authors could clarify the positioning of the work by discussing the [related work] and [scope], as suggested by Reviewer et4X.

### Self-containment

The paper could be made more self-contained by clarifying important details in the main text. For instance, explaining the implementation of the algorithm (Sec. 4.3) rather than stating that it is approximated using the scipy library. Additionally, outlining the assumptions for the regret bound (Sec. 4.5) is crucial for readers to understand the result.

### Fairness Model

The current discussion on the fairness model lacks the strength to be considered a main contribution of the paper. The fairness section introduces new assumptions, such as pushing all sensitive variables into $C^B$, in an ad-hoc manner and defers vital details to the Appendix. These results could be improved from their current preliminary state by exploring what unique properties make the suggested algorithm robust to fairness issues, delving into the implications for practical scenarios, or even adding experiments to enhance the paper.

### Incorporating the rebuttal in the main text

The AE believes that the paper underwent significant improvements during the revision period. The authors could consider reorganizing the paper more actively by adopting the feedback. For instance, the AE believes that the new experiments displaying the asymptotic behavior of regrets over a large budget could be valuable enough to be discussed in the main text. While the AE acknowledges the challenge of accommodating everything within the limited page constraints, integrating the new experiments into the original figure, without increasing space, is a viable option.

**Audience:**

Meet audience.

**Claims And Evidence:**

Well done.

**Resubmission Of Major Revision:**

The authors may consider submitting a major revision at a later time.